# Effect of maternal cigarette smoking and alcohol consumption during pregnancy on birth weight and cardiometabolic risk factors in infants, children and adolescents: a systematic review protocol

Tammy Charlene Hartel ![ORCID],[1] Eunice Bolanle Turawa,[2] André Oelofse,[1] Juléy Janice Abigail De Smidt[1]

[1]Medical Bioscience, University of the Western Cape, Bellville, Western Cape, South Africa
[2]Burden of Disease Research Unit, South African Medical Research Council, Tygerberg, South Africa

**Correspondence to**
Tammy Charlene Hartel;
3366278@myuwc.ac.za

## ABSTRACT

**Introduction** Tobacco smoking and alcohol consumption during pregnancy are particularly prevalent in low socioeconomic status populations, with an adverse association with birth outcomes and cardiometabolic risk factors. However, the direct and indirect effects of prenatal cigarette smoking and alcohol consumption during pregnancy on cardiometabolic risk in offspring have been rather inconsistent. This may be attributed to multiple factors, such as the amount and timing of exposure to tobacco smoking and alcohol during pregnancy; the influence of maternal, environmental and socioeconomic factors; or how risk factors were defined by individual researchers and studies. Therefore, this review aims to provide a summary of the most recent evidence on birth outcomes and cardiometabolic risk in children associated with alcohol and/or tobacco exposure in utero.

**Methods and analysis** PubMed, Scopus and Web of Science will be searched to identify published articles from 1 January 2001. Clinical studies that investigate the association between maternal cigarette smoking or alcohol consumption and birth weight and cardiometabolic risk factors in infants, children and adolescents will be included. Prospective cohort, case-control studies and birth cohort studies will be eligible for inclusion. Grey literature will be searched including conference proceedings, Google Scholar and the ProQuest Dissertation and Theses database. Only studies published in English will be included, with no restrictions regarding country, race or gender. Two independent reviewers will conduct the literature search and article screening. Eligibility criteria will be based on the population (infants, children, adolescents), exposure (maternal cigarette smoking, alcohol consumption or both), comparator (control group with no exposure during pregnancy) and outcomes (birth weight and cardiometabolic risk factors). Quality assessment and risk of bias will be assessed using a risk of bias tool for observational studies, and data will be extracted for analysis using a researcher-generated data extraction form. A meta-analysis will be performed to estimate pooled effect sizes if there are sufficient good-quality studies available. Sources of heterogeneity will be explored using subgroup analysis.

**Ethics and dissemination** Ethical clearance will not be required as this review will extract publicly available secondary data. Findings from this review will be disseminated via publication in a peer-review journal.

**PROSPERO registration number** CRD42021286630.

## STRENGTHS AND LIMITATIONS OF THIS STUDY

⇒ This review will synthesise literature from primary human studies that investigate the association, correlation or causation between alcohol and/or tobacco exposure during pregnancy and birth weight or cardiometabolic risk factors in offspring.

⇒ A comprehensive synthesis of all accessible data on outcomes of maternal cigarette smoking and/or alcohol consumption during pregnancy on birth weight and cardiometabolic risk factors in children will make use of a standardised risk of bias tool.

⇒ The Grading of Recommendations, Assessment, Development and Evaluations will be used to assess the quality and to strengthen the evidence from the review.

⇒ Potential publication bias might limit the review; databases will therefore be searched to find unpublished studies such as thesis dissertations and conference proceedings to minimise the risk of publication bias.

⇒ The review will be limited to evidence from an approximately two-decade period from 2001 to the final search.

## INTRODUCTION

In 2020, the global prevalence of tobacco smoking among women was 6.5%.[1] This was attributed to a substantial decline in tobacco smoking of >40% in high socioeconomic countries over the past 50 years, however,

there has been little to no decline in low-income and middle-income countries.[1] Globally, 32.4% of women 16 years and older were reported to be current alcohol consumers.[2] Despite the general decline in alcohol consumption and tobacco smoking among women globally, tobacco smoking remains a significant health burden, particularly among women of low socioeconomic status.[1] Women in low socioeconomic status communities were more likely to consume alcohol or use narcotics, concurrently with tobacco smoking during pregnancy, compared with women of higher socioeconomic status.[3–5] In addition, the global prevalence of illicit narcotic use during pregnancy was 1.6% in 2020, with studies reporting a 7.4 times higher prevalence after toxicological analysis.[3]

In South Africa, 61.2% of women reported alcohol consumption, 56.3% reported smoking tobacco only and 37.4% reported concomitant use of tobacco and alcohol during pregnancy.[4] Women of low education level, of low economic status and those that had mental health disorders were more likely to smoke during pregnancy, relapse after pregnancy and were less likely to quit smoking, compared with women with a higher education level and income status.[6] This, invariably, exacerbates the adverse effects on their offspring during and after pregnancy.[6] Similarly, biochemical analysis revealed that more than one in five pregnant women did not report their smoking status,[7] resulting in under-reported smoking and drinking patterns among pregnant women.

In the past decade, the research field of the Developmental Origins of Health and Disease (DOHaD) has significantly increased, with evidence growing on the detrimental effects of alcohol and tobacco exposure, in utero, on the development of cardiometabolic risk factors in adulthood.[8 9] Cardiometabolic risk factors include increased central adiposity, elevated triglycerides, decreased high-density lipoprotein cholesterol, elevated blood pressure and hyperglycaemia, predisposes individuals to developing type 2 diabetes mellitus and cardiovascular disease (CVD).[10] Magge et al[10] described metabolic syndrome (MetS) as having at least three of these risk factors. Despite the challenges of defining MetS in children and adolescents, Magge et al[10] emphasise that clinical screening should rather shift the focus to cardiometabolic risk factors to address the major risks associated with MetS. Moreover, cardiometabolic risk factors were reported in offspring that were exposed to tobacco smoke or alcohol in utero, which needs further investigation.[11–18] Furthermore, infants born to mothers who smoke, or those that consume alcohol during pregnancy, were reported to have an increased risk of adverse birth outcomes, such as low birth weight.[7 19] This is augmented in infants born to mothers who smoked tobacco and consumed alcohol during pregnancy compared with those born to mothers that smoked only, consumed alcohol only or abstained completely.[20] Therefore, birth weight is an important risk factor for CVD and was shown to be significantly associated with body mass index (BMI), skinfold thickness and adolescent obesity,[21 22] dyslipidaemia, altered

glucose metabolism, hypertension as well as low-grade inflammation.[23]

Whether infants are born small or large for their gestational age, both have been reported to increase their risk of overweight or obesity later in life, due to catch-up growth.[23–25] Moreover, Koklu et al[26] reported a significantly higher aortic intima-media thickness in infants that were born small for gestational age, compared with infants born appropriate for their gestational age, that was also associated with hypertriglyceridaemia.[26] Low birth weight, as well as small for gestational age was associated with higher carotid intima-media thickness values in children and linked to metabolic abnormalities, such as dyslipidaemia, abdominal obesity, hypertension and development of insulin resistance later in life.[24 27 28] Thus, overweight children or children with obesity who were born with low birth weight are more likely to present with cardiometabolic risk factors, such as elevated systolic and diastolic blood pressures, triglycerides and low-density lipoprotein cholesterol blood concentrations.[29] It is evident that tobacco smoking and alcohol use, during pregnancy, results in adverse birth outcomes which increases cardiometabolic health risks in both childhood and adulthood.[30] Therefore, the DOHaD may play a significant role in the development of cardiometabolic diseases in low-income and middle-income countries as poverty, malnutrition, licit and illicit narcotic use during pregnancy, as well as low birth weight are often prevalent in low socioeconomic regions.[12]

In summary, to help prevent CVD later in life, it is important to assess adverse birth outcomes in infants and cardiometabolic risk factors in early childhood, including low birth weight, dyslipidaemia, abdominal obesity, hypertension, high blood glucose and insulin resistance.[31] In South Africa, this is a concern as research has shown that tobacco smoking and alcohol consumption is highly prevalent in low socioeconomic populations while pregnant.[4 20 32–34] However, the evidence investigating the relationship between exposure to alcohol and tobacco during pregnancy and cardiometabolic outcomes in offspring is not consistent, warranting the present review. Therefore, the direct and indirect effects of maternal cigarette smoking and/or alcohol consumption during pregnancy on cardiometabolic risk in offspring merits further investigation. This review aims to provide a summary of up-to-date evidence on the relationship between alcohol and/or tobacco exposure in utero, birth outcomes and cardiometabolic outcomes in children.

### Justification for this review

Previous systematic reviews have addressed cardiorenal outcomes, body composition and metabolic outcomes in offspring exposed to alcohol during pregnancy,[14 35] as well as maternal smoking during pregnancy and child diabetes mellitus type 2, whereas only a few systematic reviews addressed maternal smoking during pregnancy and childhood overweight and obesity.[31 36–38] Prenatal smoking and cardiometabolic risk factors were recently

studied, but only among adults over 18 years of age.[8] Therefore, none of the latter has included all cardiometabolic risk factors in offspring and both maternal cigarette smoking and alcohol consumption during pregnancy. In addition, a review conducted in 2013 investigated maternal smoking during pregnancy and MetS in children and requires updating.[39] Therefore, a systematic review will be conducted to summarise the most recent evidence-based birth outcomes and cardiometabolic outcomes in children and adolescents in association with alcohol use and/or maternal cigarette smoking during pregnancy.

### Aim, research question and objectives
#### Aim
To provide up-to-date evidence-based birth outcomes and cardiometabolic outcomes in children associated with maternal cigarette smoking only, maternal alcohol consumption only or both maternal cigarette smoking and alcohol use during pregnancy.

#### Research question
What are the associated effects of cigarette smoking and/or alcohol consumption during pregnancy on birth weight, obesity, hypertension, diabetes, dyslipidaemia and vascular dysfunction?

#### Objectives
Primary objective: to assess birth weight and cardiometabolic risk factors associated with maternal cigarette smoking and/or alcohol consumption during pregnancy.
  Secondary objectives:
1. Identify potential mediators in the development of cardiometabolic risk factors and vascular dysfunction in offspring exposed to alcohol consumption and/or cigarette smoking during pregnancy.
2. Identify gaps in the literature and provide recommendations for future clinical studies.

### METHODS AND ANALYSIS
#### Protocol reporting and registration
The methods for this review were developed according to the Preferred Reporting Items for Systematic Reviews and Meta-Analyses Protocols (PRISMA-P) and PRISMA 2020 statement.[40 41] The systematic review protocol was registered with the international prospective register of systematic reviews (PROSPERO): CRD42021286630.

### Criteria for consideration of studies
#### Types of studies
Table 1 shows the list of inclusion and exclusion criteria. Types of studies will include human clinical studies that study the association, correlation or causation between maternal cigarette smoking and alcohol consumption and cardiometabolic risk factors in their children. Observational studies: prospective studies (cohort studies, birth cohort studies and prospective case-control studies) will be included. Data from primary population-based studies will be included only. Grey literature such as conference proceedings, theses and dessertations will be searched on the ProQuest Dissertation and Theses database, South African National Electronic Thesis and Dissertations Portal as well as Google Scholar to reduce the risk of publication bias.

#### Types of participants
Paediatric patients are defined as:
1. Infants, children and adolescents between the ages of 0 and 19 years.
2. Infants, children or adolescents exposed to maternal cigarette smoking only, maternal alcohol consumption only or both maternal cigarette smoking and alcohol during pregnancy.

#### Main outcome measures
1. Birth outcomes: low birth weight, birth weight.
2. Anthropometry: BMI, overweight, obesity, waist circumference.
3. Blood pressure: systolic blood pressure, diastolic blood pressure, hypertension, elevated blood pressure.
4. Diabetes mellitus, hyperglycaemia.
5. Dyslipidaemia, hypertriglyceridaemia, high low-density lipoprotein cholesterol, low high-density lipoprotein cholesterol.
6. Vascular dysfunction, increased intima-media thickness.

| Table 1 Inclusion and exclusion criteria | |
| --- | --- |
| **Inclusion criteria** | **Exclusion criteria** |
| ► Studies that include paediatrics (infants, children, adolescents).<br>► Observational studies: such as prospective cohort studies, birth cohort studies and prospective case-control studies will be included.<br>► Grey literature: conference proceedings, PhD dissertations and unpublished articles.<br>► Tobacco smoking in the form of cigarettes.<br>► Studies investigating the association, correlation or causation between maternal cigarette smoking and/or alcohol consumption and cardiometabolic risk factors in offspring.<br>► Studies published after 1 January 2001. | ► Studies that do not include infants, children or adolescents.<br>► Studies that do not include at least one outcome of interest will be excluded.<br>► Cross-sectional studies, retrospective case-control studies, intervention studies, clinical trials, reviews and editorials, letters and abstracts.<br>► Other forms of tobacco smoking such as electronic cigarettes, cigars, bidis, water pipes and "I quit ordinary smoking" devices. |

**Table 2** Search strategy developed in PubMed

| Search # | Search terms |
|---|---|
| #1 | maternal smoking [tiab] OR maternal smok* OR intrauterine tobacco smoke expos* [tiab] OR maternal smoking during pregnancy [tiab] OR prenatal smoking [tiab] OR prenatal smoke expos* [tiab] OR smoke pregnant [tiab] OR smoking pregnant [tiab] smoking during pregnancy [tiab] OR cigarette smoking [tiab] |
| #2 | alcohol exposure [tiab] OR maternal alcohol exposure [tiab] OR maternal alcohol expos* OR maternal alcohol consumption [tiab] OR maternal alcohol drinking OR fetal alcohol [tiab] OR foetal alcohol [tiab] OR fetal alcohol exposure [tiab] OR foetal exposure [tiab] OR alcohol exposure in utero [tiab] OR ethanol [MH] OR ethanol exposure [tiab] |
| #3 | birth outcomes [tiab] OR birth defects [tiab] OR infant, Low Birth Weight [MH] OR birth weight [MH] OR birth length [tiab] OR LBW [tiab] OR intrauterine growth restriction [tiab] OR IUGR [tiab] OR Fetal Alcohol Spectrum Disorders [MH] OR FASD [tiab] OR FASDs [tiab] |
| #4 | Metabolic syndrome [MH] OR Syndrome X [tiab] OR cardiovascular risk [tiab] OR cardiometabolic risk* [tiab] OR lipid profile [tiab] OR Lipids [MH] OR Total cholesterol [tiab] OR Triglycerides [MH] OR abnormal lipid profile [tiab] OR dyslipidemias [MH], OR dyslipidaemia [tiab] OR diabetes [tiab] OR diabetes mellitus [MH] OR blood glucose [MH] OR hyperglycemia [MH] OR hyperglycaemia [tiab] OR metabolic outcomes [tiab] OR obesity [MH] OR body composition [MH] OR Body Mass Index [MH] OR BMI [tiab] OR waist circumference [MH] OR Body Weights and Measures [MH] OR skinfold [tiab] OR anthropometry [MH] OR anthropometric measurements [tiab] OR adiposity [tiab] OR blood pressure [tiab] OR hypertens* [tiab] |
| #5 | intima media thickness [tiab] OR carotid intima media thickness [MH] OR aortic intima media thickness [tiab] OR vascular dysfunction [tiab] OR atherosclerosis [MH] OR atherosclerotic lesions [tiab] OR Plaque, Atherosclerotic [MH] OR vascular stiffness [MH] OR arterial stiffness [tiab] |
| #6 | (#1 AND #3) OR (#2 AND #3) OR (#1 AND #3 AND #4) OR (#2 AND #3 AND #4) OR (#2 AND #5 AND #3) OR (#1 AND #2 AND #3) OR (#1 AND #2 AND #4 AND #3) OR (#1 AND #2 AND #5 AND #3) |
| #7 | #6 NOT ("Animals"[Mesh] NOT "Humans"[Mesh]) |
| #8 | #7 AND "2001/01/01"[Date - Publication] : "2022/01/1"[Date - Publication] |
| #9 | Systematic Review [Publication Type] OR "Review" [Publication Type] |
| #10 | #8 NOT #9 |
| #11 | #10 AND (allchild[Filter]) |

## Search strategy

Using the comprehensive search terms, all relevant articles published, in English, from 2001, and indexed in PubMed, Google Scholar, Scopus and Web of Science will be identified for inclusion. Relevant articles will be retrieved from 2001 to gather evidence over a two-decade period. The chosen period signalled the period when significant evidence based on the current topic were published, we reckon that articles published before this period may not be relevant or of high quality to generate the robust evidence we are looking for in the current systematic review. The search strategies will incorporate both medical subject headings (MH) and free-text terms, and will be adapted to suit each database using applicable controlled vocabulary (online supplemental file 1). An experienced information specialist will review the search terms to provide inputs and to ensure that search terms are relevant and optimally sensitive to identify eligible studies. An example of the search strategy in PubMed is displayed in table 2, and in Scopus and Web of Science as shown in online supplemental tables 1 and 2, respectively. Each of the terms for exposure to maternal cigarette smoking or maternal alcohol consumption will be combined with each of the outcome terms described.

## Electronic searches

Relevant studies will be searched for and retrieved from three databases: PubMed, Scopus and Web of Science. In table 2, each of the terms for exposure to maternal cigarette smoking or alcohol consumption will be combined with each of the outcome terms described.

## Selection of studies

The initial electronic database search will be performed to obtain all eligible records. Search results for each database will be exported to Endnote, where duplicates will be removed, and the number recorded. At least two reviewers will, independently, screen the titles and abstracts for potential eligible studies and exclude irrelevant studies using the Rayyan intelligent systematic reviews tool. Disagreements will be resolved by discussion and consensus between the two reviewers or through the third reviewer's contribution. Eligibility criteria will be based on the PECO approach, an acronym representing the population, exposure, comparator and outcome.[42] P (population) of interest are infants, children and adolescents, the E (exposure) is maternal cigarette smoking, maternal alcohol consumption or both, the C (comparator) is the control group with no exposure to smoking and alcohol during pregnancy and the O (outcomes) of

interest are birth weight and cardiometabolic risk factors. Thereafter, the full text will be retrieved and exported to Endnote and reviewed independently by the two reviewers for a final selection of studies for inclusion into the review. Authors will be contacted if full texts cannot be retrieved. Experts in the field will be consulted to identify unpublished studies.

Studies will be selected if (i) relevant to the topic, (ii) study design is epidemiological observational studies such as prospective cohort studies, birth cohort studies and case-control studies, (iii) having exposure to maternal cigarette smoking or maternal alcohol consumption, (iv) having pregnant women as initially enrolled participants, (v) specified at least one adverse birth or child health outcome as an outcome of the investigation, (vi) investigated the association, correlation or causation, (vii) having the largest sample size if multiple studies used the same cohort. Only studies published in English will be included, however, there will be no restrictions regarding country, race and gender. Studies focussing on mental health, fetal alcohol syndrome, behavioural changes or cancer as the main outcome will be excluded. Any disagreement about the inclusion of studies will be resolved through the consultation of a third reviewer.

In addition to the database search, reference lists of selected articles will be examined to identify relevant studies that have not been captured in the database search. The PRISMA flow chart template will be used to summarise the search and selection of eligible studies.[43] Reasons for any exclusion of studies will be documented.

### Quality and risk of bias assessment

Quality assessment and risk of bias will be measured using the Newcastle Ottawa scale (http://www.ohri.ca/programs/clinical_epidemiology/oxford.asp) for cohort and case-control studies. Cohort and case-control studies will be awarded a maximum score of nine stars. High-quality studies will be defined as a quality score of 7 or above and a quality score below 7 will be deemed low quality.

### Data extraction and management

All data extracted from the selected studies will be captured into a Microsoft Excel document. The following data will be extracted from each selected study: authors, date of publication, country of data collection, population age at outcome assessment, sample size, study design, study setting, data collection method, type of exposure, source of data on maternal cigarette smoking/alcohol consumption, the measure of the dependent variables, birth or child health outcomes, definition of the outcome, statistical methods used to measure the outcome, adjusted covariates and effect estimates (online supplemental appendix A). Effect estimates will include relative risks (95% CI) or adjusted ORs (95% CI). Findings will include all birth outcomes or cardiometabolic outcomes tested for an association with exposure to alcohol and cigarette smoking during pregnancy. Thereafter, the data

will be extracted and stored on Review Manager (Revman Cochrane) for analyses and reporting. PRISMA guidelines will be used to summarise and report the review findings.[43]

### Data synthesis and analysis
#### Data analysis

Findings from each study will be reported as a risk ratio or an OR with the 95% CI for dichotomous variables or mean difference with the SD for continuous variables. Studies with similar types of outcomes will be grouped to obtain feasible results on an overall estimate of effect. Due to the anticipated heterogeneity in included studies, a random-effects meta-analysis will be considered.

#### Assessment of heterogeneity

Heterogeneity will be evaluated through visual inspection of forest plots to judge the extent of CI overlap. In addition, heterogeneity will be tested by the $I^2$ test. The $I^2$ statistic will classify heterogeneity. Heterogeneity among studies is expected due to risk factors defined differently according to age in children and adolescents. Therefore, the methods used to measure each health outcome are suspected to be different in children compared with adolescents. Additionally, heterogeneity may be due to characteristics such as gender, maternal health, maternal BMI and socioeconomic status. If the data allow, subgroup analysis will be performed. Similar risk factors will be grouped for comparison across studies. Risk factors according to age groups such as children or adolescents will also be grouped.

#### Meta-analysis

Where there is sufficient reporting and a low degree of heterogeneity, a meta-analysis will be performed. Quantitative data will be pooled from individual studies to determine the pooled effect estimate with a fixed model preferred. A meta-analysis will be performed using Review Manager V.5.3 (Cochrane). After the meta-analysis, the Grading of Recommendations, Assessment, Development and Evaluations (Cochrane) will be used to score and assess the overall certainty in the evidence for each outcome included in the analysis.

#### Subgroup analysis

If the data allow, subgroup analysis will be performed. Similar risk factors will be grouped for comparison across studies. Risk factors according to age groups such as children or adolescents will also be grouped. Subgroup analysis will be performed on additional risk factors which may play a role in the adverse health outcomes (such as socioeconomic disadvantage, poor nutrition, second-hand smoke exposure, ethnicity, maternal stress, maternal mental health disorders, gestational diabetes, maternal BMI, poor dietary intake, physical inactivity, adolescent substance use and tobacco smoking). The pooled effect estimate will not be determined if the source of heterogeneity cannot be explained through subgroup analysis. In addition, if a meta-analysis cannot be performed due to

few articles, insufficient reporting of data or a high degree of heterogeneity, a narrative synthesis will be reported.

## Publication bias

Begg's funnel chart will be used to perform a visual inspection and evaluation of publication bias on the selected data. This is to exclude any publication bias.

## Patient and public involvement

It was not appropriate or possible to involve patients or the public in the design, conduct, reporting or dissemination plans of our research.

## ETHICS AND DISSEMINATION

The review will extract publicly available secondary data and therefore does not require ethical review. The results will be submitted for publication in a peer-review journal.

**Correction notice** This article has been corrected since it was published. The affiliation for Eunice Bolanle Turawa has been updated.

**Contributors** TCH, JJADeS and AO conceptualised and developed the systematic review protocol. TCH wrote the first draft of the protocol. EBT and TCH developed the search strategy and EBT edited the manuscript. Both TCH and EBT will independently screen and select the title and abstracts of articles. Thereafter, the full text will be retrieved for a final review and selection. Any discrepancies regarding the eligibility of articles will be resolved through consensus and, in the event of a disagreement regarding the inclusion of articles, a consultation with a third reviewer will be held. Reasons for the exclusion of studies will be reported. TCH will extract, analyse and synthesise the data under the critical guidance of EBT. All authors will critically revise the successive drafts of the manuscript and approve the final version.

**Funding** This research was made possible through funding by the Council for Scientific and Industrial Research (CSIR).

**Competing interests** None declared.

**Patient and public involvement** Patients and/or the public were not involved in the design, or conduct, or reporting, or dissemination plans of this research.

**Patient consent for publication** Not applicable.

**Provenance and peer review** Not commissioned; externally peer reviewed.

**ORCID iD**
Tammy Charlene Hartel http://orcid.org/0000-0003-2154-5379

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
