## [Reviewer comments · BMJ Open]

ARTICLE DETAILS

TITLE (PROVISIONAL)	The effect of maternal cigarette smoking and alcohol consumption during pregnancy on birth weight and cardiometabolic risk factors in infants, children and adolescents: a systematic review protocol
AUTHORS	Hartel, Tammy; Turawa, Eunice; Oelofse, Andre; De Smidt, Juley

VERSION 1 – REVIEW

REVIEWER	Michele Bloch National Cancer Institute, Tobacco Control Research Branch
REVIEW RETURNED	20-Mar-2022

GENERAL COMMENTS	General comments to the authors: A systematic review of what is known about tobacco smoking and alcohol use on birth weight and cardio-metabolic risk factors in children and adolescents is a worthy goal. If the literature is not sufficient to draw firm conclusions in this area, especially as regards dual substance use, it will be useful to identify research needs for the future. Below, I generally limit my comments to tobacco smoking, which is my area of expertise. Early on I would recommend explaining whether the intent is to analyze the effects of combined use of alcohol and tobacco, the effects of these substances individually, or both. At present, this is unclear to the reader until the Methods section. I would also make clear what is already known on this subject – as regards tobacco smoking there is a substantial literature on the effects of prenatal smoking on infant and child health outcomes, summarized in U.S. Surgeon General's reports, among other sources. As I'm sure the authors are aware, there are diverse forms of tobacco smoking – both conventional products (e.g., cigarettes, bidis, cigars, and waterpipe use) as well as newer products (e.g., electronic cigarettes, IQOS and related products). It will be important to specify which tobacco smoking products they plan to include and exclude from their review. I suggest authors acknowledge the diverse use patterns of tobacco smoking among women around the world. Because surveillance of tobacco use during pregnancy is often lacking, tobacco use among women of reproductive age is a useful proxy. In some countries, women's tobacco smoking has consistently remained low, in others it has fallen steadily over time, and in others it remains substantial. A recent global look at women's (and men's) tobacco prevalence was published by Dai and colleagues (Dai X, Gakidou E, Lopez AD. Evolution of the global smoking epidemic over the past half century: strengthening the evidence base for policy action. Tobacco Control 2022;31:129-13). I would also consider acknowledging diverse
--

patterns of alcohol use, and note that other drugs (e.g., cannabis) may also complicate the picture, at least in some countries.

The authors are correct that in some countries, women's tobacco smoking is strongly associated with low SES. However, it is not limited to this risk factor. For example, in the U.S., prenatal tobacco smoking is strongly associated with disadvantage more generally: low income, low education, mental health and substance use, some race/ethnicities, among other factors. Disadvantage is also associated with many factors that may contribute to adverse prenatal outcomes, including poor nutrition, lack of social support, lack of access to prenatal care, stress and others. The authors may want to note how they will sort out the contribution of these other factors to adverse health in infants/children.

There are many complications of studying the effects of prenatal tobacco smoking. These include:

1. It is very difficult to disentangle the contribution of prenatal tobacco exposure from post-natal exposure to secondhand smoke (SHS) to infant/child health outcomes. Most women who smoke during pregnancy will continue to smoke after delivery, so that infants and children will experience both in utero exposure and exposure to secondhand smoke (SHS) following birth – from the mother and others in the household since tobacco smokers often “cluster” in families. In addition, many pregnant women who quit tobacco use while pregnant will resume tobacco use post-delivery.
2. Pregnant women who do not smoke themselves may be exposed to SHS from partners and others in the household. This is especially true in countries (e.g., China) where male tobacco smoking rates are high, but female tobacco use rates remain very low. I suggest authors tell the reader how they will account for SHS exposure, which is itself linked to poor infant/child health outcomes. See the 2014 U.S. Surgeon General's Report for a thorough review. Available at: https://www.cdc.gov/tobacco/data_statistics/sgr/50th-anniversary/index.htm
3. Pregnant women may be reluctant to disclose their tobacco (and alcohol) use, especially in societies where the hazards of tobacco use during pregnancy are well known and/or women's tobacco use is not socially acceptable. For example, Dietz and colleagues (Dietz et al 2010) compared self-report to biochemical verification and found that more than one in five (22.9%) U.S. pregnant women did not accurately disclose their smoking status. For this reason, absent biochemical verification, we can expect some degree of inaccuracy in pregnant women characterized as “nonsmokers” and presumably also “nondrinkers.” [See: Dietz et al 2010: American Journal of Preventive Medicine 2010; 39 (1) 45-52.]
4. Finally, some women will quit smoking upon learning they are pregnant, or later in pregnancy, and may thus be reported as non-tobacco users. However, quitting during pregnancy this will not have entirely eliminated fetal exposure in utero.

The authors plan to include cardiometabolic risk factors through adolescence, which is a time when youth may begin using either/both alcohol and tobacco; I suggest explaining how youth's own use of these substances will be considered, compared with in utero exposure.

The manuscript tends to use the term “nicotine exposure” as synonymous with tobacco smoking. This is inaccurate and should be avoided because tobacco smoking involves far more than exposure to just nicotine. Tobacco smoke contains thousands of

	chemical compounds including nicotine, carbon monoxide, carcinogens, diverse organic compounds and heavy metals; many are known or suspected to be reproductive toxicants. An excellent review of the reproductive and developmental health effects of tobacco smoking is available in the U.S. Surgeon General's Report (2010), How Tobacco Smoke Causes Disease: The Biology and Behavioral Basis for Smoking-Attributable Disease. See: https://www.cdc.gov/tobacco/data_statistics/sgr/2010/index.htm Re Page 3, line 12, which says: "...effects...have been rather inconsistent." The reader of the eventual manuscript will be interested in the authors' view on why this is so. Re Page 5 of 21, line 27, which says: "The DOHaD may be driving the increasing prevalence of non-communicable diseases in developing countries." [emphasis added]. DOHaD may play a role in eventual development of noncommunicable disease in low- and middle-income countries. However, as regards tobacco use, it is highly likely that adult use of tobacco products (which is causally associated with cancer, heart disease, lung disease, stroke and others) is likely to be a far stronger contributor. I suggest the authors adjust this language. Page 6, line 5, which says, "outcomes in children associated with teratogen exposures in utero." Because the authors are studying (only) alcohol and tobacco smoke exposure, rather than teratogens more broadly, I would be more specific. The title refers to children. Is it more accurate to refer to infants, children, and adolescents? Additional useful references are below: WHO recommendations for the prevention and management of tobacco use and second-hand smoke exposure in pregnancy, 2013. Available here: https://apps.who.int/iris/bitstream/handle/10665/94555/9789241506076_eng.pdf Collishaw, N: The Millennium Development Goals and tobacco control. Global Health Promotion 2010; 1757-9759; Supp (1): 51–59.
--	---

REVIEWER	Sabrina Luke Perinatal Services British Columbia
REVIEW RETURNED	13-Apr-2022

GENERAL COMMENTS	The study protocol investigates a timely and important issue of exposure to alcohol and smoking in utero on adverse birth outcomes and childhood cardiometabolic risk factors. The authors should consider the following revisions: 1) It is not clearly stated in the introduction if the aim of the review is to investigate the interaction between alcohol and smoking in pregnancy on outcomes or if it is to investigate either alcohol, smoking or both exposures on adverse outcomes. This needs to be clarified in the aims. The definition provided for pediatric patients states this more clearly. 2) The following statement "It is evident that tobacco and alcohol use during pregnancy results in adverse birth outcomes and influences the health of the child during both childhood and adulthood." is not supported specifically by the previous statements. Perhaps add a sentence explaining the relationship between low
---

	birth weight and future health risks in adulthood. What evidence supports this specifically? It would be beneficial to add a reference. 3) Consider defining the exposure as well. Does the smoking exposure include tobacco only or does it include cannabis or vaping for example? Is this through cigarettes only or are other routes of tobacco exposure considered?
--	--

VERSION 1 – AUTHOR RESPONSE

Reviewer 1 comments	Authors' responses
* A systematic review of what is known about tobacco smoking and alcohol use on birth weight and cardio-metabolic risk factors in children and adolescents is a worthy goal. If the literature is not sufficient to draw firm conclusions in this area, especially as regards dual substance use, it will be useful to identify research needs for the future. Below, I generally limit my comments to tobacco smoking, which is my area of expertise.	 • Thanks for commending the topic. We will identify research needs for the future if there are insufficient literature to draw firm conclusion.
*Early on I would recommend explaining whether the intent is to analyze the effects of combined use of alcohol and tobacco, the effects of these substances individually, or both. At present, this is unclear to the reader until the Methods section. *I would also make clear what is already known on this subject – as regards tobacco smoking there is a substantial literature on the effects of prenatal smoking on infant and child health outcomes, summarized in U.S. Surgeon General's reports, among other sources.	 • The aim of the systematic review is to analyze the effects of alcohol and cigarette smoking separately. In addition, where studies reported combined effect of alcohol and cigarette smoking, this will also be reported. • The effects of tobacco smoking during pregnancy and adverse health outcomes in infants and children (birth and cardiovascular health outcomes) has been added to the introduction section in the main text on page 4, line 30 to page 5, line 7.
*As I'm sure the authors are aware, there are diverse forms of tobacco smoking – both conventional products (e.g., cigarettes, bidis, cigars, and waterpipe use) as well as newer products (e.g., electronic cigarettes, IQOS and related products). It will be important to specify which tobacco smoking products they plan to include and exclude from their review.	 • The authors plan to include tobacco smoking in the form of cigarettes only and exclude other tobacco products. This has been specified and added to the inclusion and exclusion criteria on page 7, Table 1. • The term "cigarette smoking" has been added and used throughout the protocol to provide clarity.

* I suggest authors acknowledge the diverse use patterns of tobacco smoking among women around the world. Because surveillance of tobacco use during pregnancy is often lacking, tobacco use among women of reproductive age is a useful proxy. In some countries, women's tobacco smoking has consistently remained low, in others it has fallen steadily over time, and in others it remains substantial. *A recent global look at women's (and men's) tobacco prevalence was published by Dai and colleagues (Dai X, Gakidou E, Lopez AD. Evolution of the global smoking epidemic over the past half century: strengthening the evidence base for policy action. Tobacco Control 2022;31:129-13). I would also consider acknowledging diverse patterns of alcohol use, and note that other drugs (e.g., cannabis) may also complicate the picture, at least in some countries.	 • Thanks for the reference. The global prevalence of tobacco smoking and alcohol use has been added, we also acknowledged the patterns of tobacco smoking and alcohol use during pregnancy in the introduction section on page 4, line 2-18.
*The authors are correct that in some countries, women's tobacco smoking is strongly associated with low SES. However, it is not limited to this risk factor. For example, in the U.S., prenatal tobacco smoking is strongly associated with disadvantage more generally: low income, low education, mental health and substance use, some race/ethnicities, among other factors. Disadvantage is also associated with many factors that may contribute to adverse prenatal outcomes, including poor nutrition, lack of social support, lack of access to prenatal care, stress and others. The authors may want to note how they will sort out the contribution of these other factors to adverse health in infants/children.	 • The association between women's tobacco smoking and other factors such as low income, low education, mental health and substance use, ethnicity has now been added on page 4, line 12-18. The authors plan to perform a subgroup analysis on all potential confounding factors adjusted for and reported in the included studies such as socioeconomic status, low education, low income, drug/substance use, ethnicity, mental health disorders, maternal stress, poor nutrition, lack of social support and lack of access to prenatal care. This will inform the reader of the effect of these factors on adverse health outcomes in infants, children and adolescents.
There are many complications of studying the effects of prenatal tobacco smoking. These include:	

1. It is very difficult to disentangle the contribution of prenatal tobacco exposure from post-natal exposure to secondhand smoke (SHS) to infant/child health outcomes. Most women who smoke during pregnancy will continue to smoke after delivery, so that infants and children will experience both in utero exposure and exposure to secondhand smoke (SHS) following birth – from the mother and others in the household since tobacco smokers often “cluster” in families. In addition, many pregnant women who quit tobacco use while pregnant will resume tobacco use post-delivery. 2. Pregnant women who do not smoke themselves may be exposed to SHS from partners and others in the household. This is especially true in countries (e.g., China) where male tobacco smoking rates are high, but female tobacco use rates remain very low. I suggest authors tell the reader how they will account for SHS exposure, which is itself linked to poor infant/child health outcomes. See the 2014 U.S. Surgeon General’s Report for a thorough review. Available at: https://www.cdc.gov/tobacco/data_statistics/sgr/50th-anniversary/index.htm 3. Pregnant women may be reluctant to disclose their tobacco (and alcohol) use, especially in societies where the hazards of tobacco use during pregnancy are well known and/or women’s tobacco use is not socially acceptable. For example, Dietz and colleagues (Dietz et al 2010) compared self-report to biochemical verification and found that more than one in five (22.9%) U.S. pregnant women did not accurately disclose their smoking status. For this reason, absent biochemical verification, we can expect some degree of inaccuracy in pregnant women characterized as “nonsmokers” and presumably also “nondrinkers.” [See: Dietz et al 2010: American Journal of Preventive Medicine 2010; 39 (1) 45-52.] 4. Finally, some women will quit smoking upon learning they are pregnant, or later in pregnancy, and may thus be reported as non-tobacco users. However, quitting during pregnancy this will not have entirely eliminated fetal exposure in utero.	 • Although the postnatal period and childhood second-hand smoke exposure may play a role in programming during development, the systematic review will focus on maternal prenatal smoking. In cases where included studies reported the assessment of post-natal maternal smoking in the first year of life or after one years old, a subgroup analysis will be performed to measure the effect. • Thank you for the reference. Under-reporting of alcohol consumption and tobacco smoking by pregnant women has been acknowledged in the introduction section, page 4, line 14-18.
* The authors plan to include cardiometabolic risk factors through adolescence, which is a time when youth may begin using either/both alcohol and tobacco; I suggest explaining how youth’s own use of these substances will be considered, compared with in utero exposure.	 • Critical appraisal of the primary studies will be done, and studies reporting on adolescent tobacco smoking/alcohol use will be excluded in this review. The authors will conduct subgroup and sensitivity analysis. • In cases where included studies report the association between maternal prenatal smoking and

*The manuscript tends to use the term “nicotine exposure” as synonymous with tobacco smoking. This is inaccurate and should be avoided because tobacco smoking involves far more than exposure to just nicotine. Tobacco smoke contains thousands of chemical compounds including nicotine, carbon monoxide, carcinogens, diverse organic compounds and heavy metals; many are known or suspected to be reproductive toxicants.

An excellent review of the reproductive and developmental health effects of tobacco smoking is available in the U.S. Surgeon General’s Report (2010), How Tobacco Smoke Causes Disease: The Biology and Behavioral Basis for Smoking-Attributable Disease. See: https://www.cdc.gov/tobacco/data_statistics/sgr/2010/index.htm

*Re Page 3, line 12, which says: “...effects...have been rather inconsistent.” The reader of the eventual manuscript will be interested in the authors’ view on why this is so.

*Re Page 5 of 21, line 27, which says: “The DOHaD may be driving the increasing prevalence of non-communicable diseases in developing countries.” [emphasis added]. DOHaD may play a role in eventual development of noncommunicable disease in low- and middle-income countries. However, as regards tobacco use, it is highly likely that adult use of tobacco products (which is causally associated with cancer, heart disease, lung disease, stroke and others) is likely to be a far stronger contributor. I suggest the authors adjust this language.

*Page 6, line 5, which says, “outcomes in children associated with teratogen exposures in utero.” Because the authors are studying (only) alcohol and tobacco smoke exposure, rather than teratogens more broadly, I would be more specific.

*The title refers to children. Is it more accurate to refer to infants, children, and adolescents?

Additional useful references are below:
WHO recommendations for the prevention and management of

cardiometabolic risk factors after adjusting for youth’s own use of alcohol and tobacco smoking, a subgroup analysis will be performed.

- The term “nicotine exposure” has been replaced with “tobacco smoking” throughout the manuscript.

- The following sentence was added into the abstract: “This may be attributed to the amount of exposure, timing of exposure, influence of maternal, environmental and socioeconomic factors, or how risk factors may be defined.” Any factors identified will be further discussed in the systematic review.

- The authors have changed the sentence ““The DOHaD may be driving the increasing prevalence of non-communicable diseases in developing countries.” to “Therefore, the DOHaD may play a role in the development of cardiometabolic diseases in developing countries as poverty, malnutrition, licit and illicit drug use during pregnancy and

tobacco use and second-hand smoke exposure in pregnancy, 2013. Available here: https://apps.who.int/iris/bitstream/handle/10665/94555/9789241506076_eng.pdf Collishaw, N: The Millennium Development Goals and tobacco control. Global Health Promotion 2010; 1757-9759; Supp (1): 51–59	low birth weight are often prevalent in the same regions.”  • The term “teratogens” has been changed to “alcohol consumption and cigarette smoking” to be more specific. • The authors changed the title to “The effect of maternal smoking and alcohol consumption during pregnancy on birth weight and cardio-metabolic risk factors in infants, children and adolescents: A systematic review protocol”. • Thanks for the additional references.
Reviewer: 2 Comments	Authors responses

* The study protocol investigates a timely and important issue of exposure to alcohol and smoking in utero on adverse birth outcomes and childhood cardiometabolic risk factors. The authors should consider the following revisions.	 • Thanks for your suggestions/recommendations.
1) It is not clearly stated in the introduction if the aim of the review is to investigate the interaction between alcohol and smoking in pregnancy on outcomes or if it is to investigate either alcohol, smoking or both exposures on adverse outcomes. This needs to be clarified in the aims. The definition provided for pediatric patients states this more clearly.	 • The aim of the systematic review is to investigate the effects of alcohol only and tobacco only. The authors will analyse both alcohol and tobacco smoking, provided that studies have included the exposure to both alcohol and tobacco. This has been stated more clearly in the Aim on page 6, line 18-21.
2) The following statement "It is evident that tobacco and alcohol use during pregnancy results in adverse birth outcomes and influences the health of the child during both childhood and adulthood." is not supported specifically by the previous statements. Perhaps add a sentence explaining the relationship between low birth weight and future health risks in adulthood. What evidence supports this specifically? It would be beneficial to add a reference.	 • The relationship between low birth weight and cardiovascular health risks later in life has been added in the introduction section on page 5, line 8-22.
3) Consider defining the exposure as well. Does the smoking exposure include tobacco only or does it include cannabis or vaping for example? Is this through cigarettes only or are other routes of tobacco exposure considered?	 • The authors plan to include tobacco smoking in the form of cigarettes only and exclude other tobacco products. This has been added to the inclusion and exclusion criteria on page 7, table 1.

VERSION 2 – REVIEW

REVIEWER	Michele Bloch National Cancer Institute, Tobacco Control Research Branch
REVIEW RETURNED	03-Jun-2022

GENERAL COMMENTS	Dear Author Group, Thank you for the opportunity to review your systematic review protocol, which has improved in many ways since the earlier submission. Below are a few additional comments to consider. The overall standard of English has improved, but an additional review for English style/grammar would be beneficial Throughout, I recommend that each sentence that makes a comparison specify the comparator group. For example, this sentence on page 4, lines 16-18, does not provide the comparison group: “These women are more likely to consume alcohol or use drugs....”
---

	Throughout, ensure that any sentence that provides prevalence data (e.g., for alcohol, tobacco or other drug use) indicates whether the statistics refers to the global level, regional level, for low-income countries, for a specific country, or other. This is because use patterns vary widely by region and country. For example, this sentence on page 4, lines 20-21: “To emphasize, in low socioeconomic populations an alarming 61.2% of women consume alcohol, 56.3% smoke tobacco only, and 37.4% consume alcohol only [4]. Note: In addition, this sentence is confusing to the reader as written. Page 5, lines 43-45: I would recommend editing the current sentence to read: “In summary, to help prevent CVD later in life....” “Help prevent” is needed because prenatal factors are not the only important factors in development of CVD in later life. Page 6, lines 44-45. Consider whether your research question should read: “What are the associated effects of cigarette smoking and/or alcohol consumption during...” because you are looking at these factors alone and in combination. If you decide to use an alternative formulation, please use it consistently throughout the manuscript. Page 10, lines 58-59. Fetal alcohol syndrome is not classified as a mental health disorder.
--	---

REVIEWER	Sabrina Luke Perinatal Services British Columbia
REVIEW RETURNED	24-May-2022

GENERAL COMMENTS	Thank you for addressing the reviewer comments.
---

VERSION 2 – AUTHOR RESPONSE

Reviewer 1 comments	
The overall standard of English has improved, but an additional review for English style/grammar would be beneficial	 • Thank you. The standard of English, grammar and spelling have been checked by an additional proof-reader.
Throughout, I recommend that each sentence that makes a comparison specify the comparator group. For example, this sentence on page 4, lines 16-18, does not provide the comparison group: “These women are more likely to consume alcohol or use drugs....”	 • A comparison group has been added to this sentence, checked and added throughout the manuscript.
Throughout, ensure that any sentence that provides prevalence data (e.g., for alcohol, tobacco or other drug use) indicates whether the statistics refers to the global level, regional level, for low-income countries, for a specific country, or other. This is because use patterns vary widely by region and country. For example, this sentence on page 4,	 • The global, regional or country-level has been specified throughout and changes made to indicate the level of discussion. • Thank you. The sentence has been rephrased for clarity: “To emphasise, in South Africa, 61.2% of women consume

lines 20-21: "To emphasize, in low socioeconomic populations an alarming 61.2% of women consume alcohol, 56.3% smoke tobacco only, and 37.4% consume alcohol only [4]. Note: In addition, this sentence is confusing to the reader as written.	alcohol, 56.3% smoke tobacco only and 37.4% reported concomitant use of tobacco and alcohol during pregnancy." Now on page 4, lines 13-14.
Page 5, lines 43-45: I would recommend editing the current sentence to read: "In summary, to help prevent CVD later in life...." "Help prevent" is needed because prenatal factors are not the only important factors in development of CVD in later life.	 • Thank you for the suggestion. This part of the sentence has been changed.
Page 6, lines 44-45. Consider whether your research question should read: "What are the associated effects of cigarette smoking and/or alcohol consumption during..." because you are looking at these factors alone and in combination. If you decide to use an alternative formulation, please use it consistently throughout the manuscript.	 • The research question has been changed to "What are the associated effects of cigarette smoking and/or alcohol consumption during pregnancy on birth weight, obesity, hypertension, diabetes, dyslipidaemia, and vascular dysfunction?"
Page 10, lines 58-59. Fetal alcohol syndrome is not classified as a mental health disorder.	 • Thank you. Fetal alcohol syndrome has been separated from mental health.

VERSION 3 – REVIEW

REVIEWER	Michele Bloch National Cancer Institute, Tobacco Control Research Branch
REVIEW RETURNED	25-Jun-2022

GENERAL COMMENTS	The standard of English continues to improve. The authors have been responsive to previous suggestions. Please note: Table 1 refers to IQOS and indicates this is an acronym for "I Quit Original Smoking." Per WHO, the appropriate acronym for IQOS is "I Quit Ordinary Smoking," although the manufacturer (PMI) claims that IQOS is not an acronym. My suggestion is to trust the WHO information. Best wishes as you move forward to conduct your systematic review.
--